# The Role of TRPC1 in Modulating Cancer Progression

**DOI:** 10.3390/cells9020388

**Published:** 2020-02-07

**Authors:** Osama M Elzamzamy, Reinhold Penner, Lori A Hazlehurst

**Affiliations:** 1Clinical and Translational Sciences Institute, School of Medicine, West Virginia University, Morgantown, WV 26506, USA; omelzamzamy@mix.wvu.edu; 2The Queen’s Medical Center and University of Hawaii, Honolulu, HI 96813, USA; rpenner@hawaii.edu; 3Pharmaceutical Sciences, School of Pharmacy and WVU Cancer Institute, West Virginia University, Morganton, WV 26506, USA

**Keywords:** TRPC1, SOCE, cancer progression, EMT

## Abstract

Calcium ions (Ca^2+^) play an important role as second messengers in regulating a plethora of physiological and pathological processes, including the progression of cancer. Several selective and non-selective Ca^2+^-permeable ion channels are implicated in mediating Ca^2+^ signaling in cancer cells. In this review, we are focusing on TRPC1, a member of the TRP protein superfamily and a potential modulator of store-operated Ca^2+^ entry (SOCE) pathways. While TRPC1 is ubiquitously expressed in most tissues, its dysregulated activity may contribute to the hallmarks of various types of cancers, including breast cancer, pancreatic cancer, glioblastoma multiforme, lung cancer, hepatic cancer, multiple myeloma, and thyroid cancer. A range of pharmacological and genetic tools have been developed to address the functional role of TRPC1 in cancer. Interestingly, the unique role of TRPC1 has elevated this channel as a promising target for modulation both in terms of pharmacological inhibition leading to suppression of tumor growth and metastasis, as well as for agonistic strategies eliciting Ca^2+^ overload and cell death in aggressive metastatic tumor cells.

## 1. Introduction

Calcium is a ubiquitous second messenger that is known to regulate a myriad of physiological cellular functions in normal cells [1]. Levels of intracellular free concentration of Ca^2+^ is tightly regulated with strict spatial and temporal control to initiate, maintain, and terminate appropriate signaling pathways and phenotypes. Changes in intracellular Ca^2+^ concentration include fast processes that require milliseconds of intracellular Ca^2+^ spikes necessary for exocytosis and muscle contraction, to processes requiring minutes to hours of Ca^2+^ flux that affect cellular proliferation, cell cycle control, migration, gene expression, and cell death [2,3]. Ion channels play a fundamental role in defining regulatory signaling pathways in the progression of cancer. Further, numerous studies indicate that Ca^2+^-dependent signaling pathways are involved in augmenting tumor proliferation, differentiation, migration, invasion, metastasis, and apoptosis, thus tumor cells often exhibit increased expression of Ca^2+^ regulatory networks [2,3,4]. Moreover, given the importance of Ca^2+^ signaling, it is not surprising that cells tightly and precisely regulate proteins handling Ca^2+^ signals, including receptors, channels, and transporters [5]. Regulation of Ca^2+^ homeostasis includes transient Ca^2+^ release from intracellular stores (ER/SR, mitochondria, lysosomes) as well as more sustained influx of extracellular Ca^2+^ [6]. The store-operated Ca^2+^ entry pathway (SOCE) is a major Ca^2+^ entry pathway in non-excitable cells and SOCE activity is known to modulate insensitivity to antigrowth signals via multiple routes, as reviewed by Prevarskaya et al. [4]. There are numerous channels and transporters that regulate Ca^2+^ levels, and deciphering the role and interplay of individual members in facilitating tumor progression remains challenging. There is a great deal of controversy surrounding the role of TRPC1; while some reports suggest it is an ion channel, other reports suggest that TRPC1 alone is not sufficient to form an ion channel and functions as a modulator of other TRPC channels. Additionally, the expression of TRPC1 as a prognostic marker in cancer appears to be context specific as TRPC1 expression has been reported to be associated with poor clinical outcomes for certain types of cancers, while in other indications it is reported to be associated with improved outcomes. This review will focus on the role of TRPC1 expression as a determinant of SOCE and cancer progression.

The transient receptor potential (TRP) ion channels were first discovered in *Drosophila melanogaster* by studying photo-transduction [7]. The TRP protein superfamily shares similarities in structure to the parent *Drosophila* TRP and were initially classified into three subfamilies TRP-Canonical, TRP-Vanilloid, and TRP-Melastatin (TRPC, TRPV, and TRPM, respectively) [8]. Later, the TRP superfamily was classified into seven subfamilies; TRP-Classical/Canonical (TRPC), TRP-Vanilloid (TRPV), TRP-Melastatin (TRPM), TRP-Ankyrin (TRPA), TRP-Polycystin (TRPP), and TRP-Mucolipin (TRPML). The non-mechanoreceptor potential C-TRP (TRPN) is comprised of approximately 30 members [9]. Except for TRPM4 and TRPM5, which are Ca^2+^-activated monovalent-selective cation channels [10,11], TRP family members are non-selective channels that are permeable to Ca^2+^ to varying degrees [9]. TRP channels generally share structural similarities that include six-transmembrane domains, and the proteins typically assemble as homotetrameric or in some cases heterotetrameric channels summarized by Strubing and colleagues [12].

In addition to TRP channels, the SOCE mechanism of action is dependent on the depletion of the endoplasmic reticulum (ER) Ca^2+^ stores through ryanodine receptors (RyRs) or inositol 1,4,5-trisphosphate receptors (IP_3_R) [13,14]. SOCE is regulated by agonist binding surface receptors, including G-protein coupled receptors (GPCRs) or receptor tyrosine-kinases (RTKs), activating phospholipase Cβ (PLCβ) via Gq/11 and PLCγ via RTK-mediated signaling [2,6]. This results in the enzymatic cleavage of plasma-membrane phosphatidylinositol 4,5-bisphosphate (PIP_2_) into IP_3_ and diacylglycerol (DAG). The depletion of Ca^2+^ stores from the ER is sensed by the transmembrane protein stromal interaction molecules (STIM1 and STIM2), as Ca^2+^ dissociates from the EF domain of STIM1 and/or STIM2 [15]. STIM molecules multimerize and translocate to ER–PM junction to form puncta that co-assemble with any or all of three calcium-release-activated calcium (CRAC) channel subunits ORAI1/2/3. This protein–protein interaction between STIM and ORAI results in the sustained opening of the highly Ca^2+^-selective CRAC channels that allow for both cytosolic Ca^2+^ signaling and replenishing of ER stores [6]. Additionally, in some cell types, STIM1 may intersect with ORAI1 and members of the TRPC subfamily by its reported capability to directly interact with TRPC1, TRPC4, and TRPC5, and indirectly with TRPC3 and TRPC6 (Figure 1A) [16,17,18,19].

The TRPC subfamily consists of seven members (TRPC1-7), and they are known to function as non-selective cation channels, with permeability to Ca^2+^, Na^+^, and K^+^ [20]. The role of TRPC1 in SOCE activity has been discussed in a recent report by Dyrda and colleagues, where they reported that TRPC1 activation is dependent on activation of the Icrac current activated by STIM1 and comprised of ORAI1/2/3 [21]. However, activation of STIM1 does not necessarily activate TRPC1, as there are two proposed mechanisms for the store-operated channels activation. The transmembrane protein STIM1 interacts with ORAI1 activating the CRAC channels, with Ca^2+^ selective Icrac currents [22,23,24,25]. STIM1 interacts with TRPC1, forming the STIM-ORAI1-TRPC1 complex and activating the SOC channels conducting cation non-selective Isoc currents [25,26]. This experimental evidence supports a model in which, following the activation of the SOC channels, the non-selective cation channels TRPC1, TRPC4, and TRPC5 form heterotetramers, as TRPC1-TRPC4, TRPC1-TRPC5, or TRPC1-TRPC4-TRPC5 become operative to facilitate further Ca^2+^ entry (Figure 1B) [27,28,29]. TRPC1 has been under study in recent years as a Ca^2+^ channel in the SOCE pathway, on the other hand, recent reports have suggested that TRPC1, when expressed on it is own, is not sufficient to form a channel. The role of TRPC1 has been studied extensively in recent years to investigate its function as a calcium channel or a modulator for other TRPC channels. Despite all data and studies into TRPC proteins, their mechanism of action remains poorly understood. Some members of the TRPC subfamily are capable of forming channels when expressed alone by forming homomers (TRPC4-TRPC4 and TRPC5-TRPC5) [30]. However, Storch et al. reported that TRPC1 is incapable of forming a Ca^2+^-permeable channel by itself, but is essential in forming heteromers with all other members of the TRPC subfamily [31]. Further, they reported a decrease in Ca^2+^ permeability in heteromeric complexes containing TRPC1, but the effect of TRPC1 in Ca^2+^ entry and as a member of the SOCE pathway is tissue dependent.

The role of Ca^2+^ signaling through the SOCE pathway involving STIM1 and ORAI1 has been reported previously to affect the prognosis of cervical, colorectal, breast, esophageal, multiple myeloma, and lung cancers by affecting tumor growth, proliferation, metastasis, and survival [3,32,33,34]. In this review we aim to address the role of TRPC1 as a member of the SOCE pathway in promoting the advancement of the hallmarks of cancer, and consider their potential as therapeutic targets for developing novel cancer treatments.

## 2. Ca^2+^ Signaling Through SOCE Modulates Gene Expression

Considering the numerous pathways that Ca^2+^ signaling modulates, including kinases, phosphatases, proteases, and metabolic enzymes, the role of Ca^2+^ signaling in the progression of cancer is likely multi-factorial. However, one attractive Ca^2+^-dependent candidate pathway is the activation of NFAT (reviewed by Putney [35]). Increased intracellular Ca^2+^ levels occur via multiple channels, but the increased oscillatory Ca^2+^ currents through emptying of intracellular stores, and extracellular Ca^2+^ flux through the SOCE is the major contributor to Ca^2+^ -regulated gene expressions [36]. Ca^2+^ binding to its receptor calmodulin activates the phosphatase protein calcineurin, which in turn dephosphorylates NFAT, leading to its translocation to the nucleus. NFAT is a transcription factor known to regulate the expression of genes encoding cytokines and receptors mandatory for T-cell survival. Multiple reports indicate that tumor cells have hijacked the calcium-dependent NFAT pathway to support cytokine-dependent survival and homing. For example, in diffuse large B-cell lymphoma (DLCBCL), Bucher et al. reported that NFAT activity is chronically elevated in tumor cells; a finding that correlated with elevated Ca^2+^ levels. In addition, inhibition of calcineurin with cyclosporin A or FK506 treatment reduced NFAT target genes including EGR2, IL10, NFKB1A, and Jun, and induced cell death in DLCBCL cell lines [37]. Similarly, Urso et al. showed that NFATc3 is a critical determinant of proliferation and migration of U251 glioblastoma cell lines [38]. The authors demonstrated that reducing the expression of NFAT3c inhibited the expression of TNF-alpha, GM-CSF, IL-2, and CXCR-3 when U251 cells were treated with an ionophore. Finally, the authors demonstrated that a reduction of NFAT3c inhibited the growth of U251 cells in vivo. The role of NFAT in cancer progression has been comprehensively reviewed by Mancini et al., where they reported that NFAT overexpression in solid and hematological tumors plays a role in tumor survival, migration, and invasion [39] (Figure 2).

## 3. Pharmacological and Genetic Tools Used to Probe the Role of TRPC in Cancer

Numerous studies have addressed the role of TRPC1 as a Ca^2+^ channel using siRNA or shRNA strategies to reduce the expression of TRPC1 [31,40,41,42,43,44,45,46,47,48,49,50,51,52,53]. The outcomes of silencing TRPC1 in cancer cells are addressed below in detail in context to disease indication and contribution to growth and metastasis, and summarized in Table 1. Other studies have employed various pharmacological tools to determine the role of TRPC1 along with TRPC4 and TRPC5 as contributors to the SOCE pathways and Ca^2+^ entry. Some of these tools are inhibitors that suppress SOCE and cell proliferation, including 2-APB [40,43,44,48,49,51,52,53,54,55,56], SKF96365 [31,51,53,55,57], and MRS1845 [51,53], whereas the plant-derived sesquiterpenoid englerin A can be used as an agonist for TRPC4 and TRPC5 to promote Ca^2+^ overload and cell death [41,58]. 2-APB was originally discovered as an IP3R inhibitor, but has since been recognized as a fairly non-selective inhibitor of a large number of TRP channels that lack specificity for TRPC1 [59,60]. Similarly, SKF96365 was originally discovered as an inhibitor of the receptor-mediated Ca^2+^ entry in platelets and endothelial cells. The compound is also pleiotropic and found to inhibit SOCE via STIM1, to block various TRPC channels, as well as voltage-gated K^+^ and Ca^2+^ channels [61,62]. Finally, the dihydropyridine MRS1845 has been reported as an inhibitor for the SOCE pathway in the HL-60 leukemia cell line [63], but its potential direct effects on TRPC1 have not yet been established in electrophysiological studies. Thus, these inhibitors cannot be considered as specific reagents for TRPC1, and genetic confirmation is required when using these pharmacological tools. Other more specific antagonists for TRPC1, TRPC4, and TRPC5 have been discussed by Rubaiy, and include PICO 145, Clemizole, M084, AC1903, Galangin, and AM12. These tools could facilitate future studies aimed at better understanding the role of TRPC1 in cancer [64].

## 4. TRPC1 Expression and Correlation With Proliferation, EMT, and Migration

### 4.1. Pancreatic Cancer

Epithelial-mesenchymal transition (EMT) is a known modulator and a key step into tumor invasion and metastasis. One of the key modulators of EMT is TGF-β, which has been reported previously to induce EMT in mammary epithelial cells [65]. The role of TRPC1 as a Ca^2+^ channel in pancreatic cancer cell proliferation and its development was proposed as being a downstream effector of TGFβ signaling [52,56]. In SMAD4-null pancreatic cancer cells, TGFβ is reported to induce a cytosolic Ca^2+^ increase, leading to the activation of the Ca^2+^-dependent protein kinase C-α (PKCα) and its translocation to the plasma membrane. Further, TGFβ activated PKCα-dependent cellular motility and migration, by inhibiting the tumor suppressor PTEN [56]. Later, TRPC1 4 and 6 levels were shown to be high in pancreatic cancer cells, indicating their function as TGFβ mediators for Ca^2+^ entry [52]. Pharmacologically inhibiting SOCE pathways using 2-APB and La^3+^ abrogated the TGFβ-dependent increase in cytosolic Ca^2+^ levels. Blocking PKCα by the selective PKCα inhibitor Gö-6976 also inhibited TGFβ-induced Ca^2+^ entry [52]. Further, siRNA knockdown of TRPC1 or treatment with 2-APB significantly inhibited pancreatic cancer cell motility induced by TGFβ, although, interestingly, siRNA knockdown of TRPC4 and 6 had no effect on TGFβ-induced pancreatic cell motility [52].

### 4.2. Breast Cancer Epithelial-Mesenchymal Transition and Proliferation

It has previously been reported that epithelial–mesenchymal transition (EMT) is associated with upregulation of SOCE via increased levels of STIM1 and ORAI1 in MCF7 and MDA-MB-231 cell lines, promoting invasion and proliferation of breast cancer cells [54]. However, the role of the SOCE pathway in EMT may be dependent on the stimulus inducing the transition and/or cell context specific. For example, MDA-MB-468 cell lines undergoing epidermal growth factor (EGF)-induced EMT correlate with a reduction of SOCE activity, and the reduction in Ca^2+^ flux was associated with ORAI1 downregulation, while TRPC1 expression was not altered [50]. Interestingly, hypoxia-induced EMT increased the expression of TRPC1 in MDA-MB-468, MDA-MB-231, and HCC1569 cell lines, an effect that required HIF1α expression, indicating that TRPC1 is a HIF1α target. Reducing TRPC1 expression inhibited hypoxia-induced increased Snail, Vimentin, and Twist expression. Reducing the expression of TRPC1 in hypoxia resulted in decreased basal Ca^2+^ levels, but increased SOCE activity was noted by depleting intracellular stores using sarco/endoplasmic reticulum Ca^2+^ ATPase (SERCA) inhibitor cyclopiazonic acid [42]. Reducing the expression of TRPC1 in MDA-MB-468 cells decreased the proliferation rate and was associated with reduction of cells in the S-phase, while ORAI1 had no effect [50].

MCF7 breast cancer cells proliferate in response to activation of the Ca^2+^-sensing receptor (CaR) by extracellular Ca^2+^ or its agonist spermine [66]. El Hiani and colleagues reported that Ca^2+^-mediated CaR activation in MCF7 cells results in activation of PLC and PKC [55]. They further reported that proliferation is dependent on activation of ERK1/2, which was shown to be activated downstream of PLC and PKC. Reducing the expression of TRPC1 attenuated ERK1/2 phosphorylation mediated by CaR activation, which is necessary for CaR-induced cell proliferation of MCF7 cells. In addition, reducing TRPC1 expression inhibited MCF7 proliferation by halting the cell cycle progression at the G1 phase [45]. Cell cycle progression was dependent on TRPC1 mediating the activity of Ca^2+^-activated K^+^ channels (KCa3.1) [45,55]. Interestingly, a feed-forward loop was described where TRPC1 expression was dependent on activation of EGFR and ERK activity in MCF7 cells [67]. In human breast ductal adenocarcinoma primary patient samples, TRPC1, TRPC6, TRPM7, and TRPM8 were reported to be overexpressed in cancer cells compared to normal adjacent tissue. Importantly, increased TRPC1 expression correlated with expression and increased proliferative and invasive capacity of small grade I breast cancer tumors [68]. Taken together, these data suggest that TRPC1 expression may play a role in facilitating EMT and proliferation, and further studies are required to carefully delineate the mechanism underpinning the role of TRPC1 in mediating EMT and proliferation in breast cancer.

### 4.3. Glioblastoma

In D54MG cells, a model for malignant gliomas or glioblastoma multiforme (GBM), the pharmacological inhibition of SOCE with 2-APB, SKF96365, and MRS1845 significantly reduced both Ca^2+^ influx and proliferation as well as the formation of multinucleated cells, a characteristic of GBM [53]. To define the role of TRPC1 as a Ca^2+^ channel involved in SOCE, its function was blocked using a polyclonal-TRPC1 antibody, which reduced the Ca^2+^ entry by 25%, whereas no effect was seen with TRPC5 inhibition [53]. Similarly, Ca^2+^ entry was significantly decreased following the reduction of TRPC1 expression using shRNA [53]. The knockdown of TRPC1 also significantly reduced proliferation and resulted in enlarged multinucleated cells. Moreover, TRPC1 expression levels were reportedly down-regulated in patients with giant cell glioblastoma [53].

It has been reported that lipid rafts microdomains (LRDs) and plasma membrane caveolin-1 are critical for TRPC1 insertion into the plasma membrane and its activity as a Ca^2+^ channel, as TRPC channels contain a caveolin-1 binding domain [69,70,71]. Indeed, TRPC1 was implicated in chemotaxis and directional migration of D54MG cells towards epidermal growth factor (EGF), by co-localizing with caveolin-1 and lipid rafts in the cells’ leading edge, which was abrogated with inducible TRPC1 shRNA knockdown [51]. Consistent with the importance of Ca^2+^ signaling, the use of SOCE inhibitors MRS1845 and SKF96365 disrupted glioma migration [51]. Evidence that TRPC1 expression is essential for growth in vivo was assessed by injecting nude mice glioma cells ectopically expressing doxycycline-inducible shRNA targeting TRPC1. Mice with inducible knockdown of TRPC1 had a shallow, significantly smaller tumor size compared to wild-type TRPC1 expression. However, propensity for metastasis was not evaluated in these mice [53].

### 4.4. Lung Cancer

Lung cancer is the leading cause of death from cancer in men and women and thus clinical data continue to indicate the need for new treatment strategies to improve patient outcomes [72]. Levels of TRPC protein expression and their relation to tumor prognosis have been reported previously and identified as a potential target for treatment. TRPC1, TRPC3, TRPC4, and TRPC6 levels have been found to be highly expressed in patient specimens compared to other TRPC channels [49,73]. Moreover, overexpressing TRPC1 and TRPC6 in the A549 NSCLC cell line was sufficient to increase proliferation, while blocking TRPC channels by the IP_3_ receptor inhibitor and SOCE modulator 2-APB, or the specific TRPC1 antibody T1E3, inhibited proliferation of A549 cells [49].

As mentioned earlier, TRPC1 levels in breast cancer were dependent on HIF-1α following hypoxia-induced EMT. Wang et al. reported similar results in lung cancer by exposing A549 cells to high nicotine levels, which in turn resulted in increased HIF-1α levels (as shown previously by Guo et al. [74]), leading to the upregulation of SOCE components, namely ORAI1, TRPC1, and TRPC6 [75]. Further, the nicotine exposure was associated with increased basal intracellular Ca^2+^ levels in A549 cells related to constitutive SOCE activity. Downregulating HIF-1α was associated with low expression of ORAI1, TRPC1, and TRPC6, and resulted in decreased proliferation of A549 cells. Silencing TRPC1 decreased hypoxia-induced autophagy [75]. Autophagy is known to promote survival in hypoxic environments [76], albeit no data was reported on survival with respect to exposure to hypoxia and TRPC1 expression. More recently, STIM1 and TRPC1 were shown to mediate cisplatin cytotoxicity in NSCLC A549 cell line by facilitating the DNA damage response (DDR) and reactive oxygen species (ROS) production leading to apoptosis. While these effects were discovered mostly by silencing STIM1, and thus do not rule out ORAI1 or other SOCE components, the overall effect was mainly mediated by inhibition of Ca^2+^ influx through the SOCE pathway [40].

### 4.5. Colon Cancer

In colon cancer, the role of SOCE members, mainly TRPC1 and ORAI1, has been discussed thoroughly by Villalobos et al. [77]. Basal Ca^2+^ levels have been shown to be higher in HT29 colon carcinoma cell lines with higher increase in cytosolic Ca^2+^ in response to agonists like ATP and carbachol, compared to the normal human mucosa cell line NCM460 [48]. Although TRPC1 mRNA levels were similar to other SOCE members in HT29 cells, there was an increase in protein expression levels compared to others, and TRPC1 silencing was associated with decreased store-operated (Isoc) currents. As previously mentioned, TRPC1 facilitates the migration of GBM cells towards EGF. Guéguinou et al. investigated the mechanism by which TRPC1 contributes to colon cancer cell migration. They demonstrate that reducing the expression of TRPC1 inhibits migration of the HCT-116 colon cancer cell line by disrupting the complex formation of Ca^2+^-activated K^+^ channels (SK3), ORAI1, and TRPC1 in the lipid rafts [44]. Further, colon cancer cell migration was shown to be dependent on EGFR activation, leading to downstream activation of the PI3K/Akt pathway. This promotes the phosphorylation activation of STIM1 and SOCE, leading to the translocation of TRPC1 and ORAI1 in the lipid rafts to form a complex with SK3, which allows a loop formation of further Akt activation and TRPC1-ORA1-SK3-dependent migration. Taken together, TRPC1 expression appears to have a role in the migration of multiple cancer types and more studies are required to determine whether inhibition of this pathway will block metastasis of primary tumors.

**Table 1 cells-09-00388-t001:** Role of TRPC expression or activity in augmenting proliferation and metastasis in cancer.

*Cancer Type*	*Cell Type*	*Native TRPC Expression*	*Silenced Proteins/ Tools Used*	*Native Expression Effect /Silencing Effect*	*Reference*
*Pancreatic*	BxPc3 (human ductal adenocarcinoma)	↑ TRPC1	SOC/2-APB TRPC1/siRNA	Motility/ ↓ motility (TGFβ-dependent motility)	[52]
*Breast*	MDA-MB-468 (EGF-mediated EMT cells) (human breast adenocarcinoma)	Comparable to MDA-MB-231 - EMT	TRPC1/siRNA	/↓ Cell proliferation (↓S-phase)	[50]
	MDA-MB-468 hypoxia-mediated EMT cells)	↑ TRPC1 and TRPC3	TRPC1/siRNA	/↑ Ca^2+^ influx in SOCE and ↓ autophagy marker LC3BIII	[42]
	MCF7 (adenocarcinoma)	↑TRPC1	TRPC1/siRNA	Proliferation/↓ proliferation (↓G1-phase)	[45,55]
	Primary patient TNBC cells (mesenchymal subtype)	↑ TRPC1	-	Worsened prognosis/ -	[42]
	Primary human breast ductal adenocarcinoma	↑ TRPC1 ↑ TRPC6	-	↑ proliferation and invasion/-	[68]
*Glioblastoma Multiforme*	D54MG (GMB Cell line)	-	TRPC1/2-APB, SKF96365, MRS1845, polyclonal TRPC1 antibody, and shRNA	Proliferation, migration/ ↓ Ca^2+^ influx in SOCE, ↓ proliferation	[51,53]
*Lung*	Primary patient cells (NSCLC)	↑ TRPC1, 3,4,6	-	High expression with well-differentiated tumor	[49,73]
	A549 (NSCLC cell line)	↑ TRPC1 ↑ TRPC6	SOC/2-APB TRPC1, TRPC3-TRPC6/T1E3, and T3667E3 ab	↑ Proliferation/↓ proliferation	[49]
	A549 (hypoxia-mediated EMT by nicotine treatment)	↑TRPC1 ↑TRPC6, and ↑ORAI1	↓TRPC1/siRNA HIF-1α	↑ SOCE activity/↓ proliferation, ↓hypoxia-induced autophagy	[75]
*Colon*	HT29 (human colon carcinoma)	↑ TRPC1 (protein)	SOC/2-APB TRPC1/siRNA	↑ SOCE, ↑ proliferation/↓ Isoc currents, ↓ invasion	[48]
	HCT116	-	TRPC/siRNA	Migration/↓ migration	[44]

(-) Indicates no available data.

## 5. Activation of the SOCE Pathway for Inducing Cell Death in Cancer

Cancer requires robust Ca^2+^ signaling to support proliferation, invasion, and metastasis, whereas excessive Ca^2+^ levels lead to apoptosis and cell death; both the suppression and enhancement of the SOCE pathways may be exploited for treatment of cancer. In triple-negative breast cancer (TNBC), Grant et al. reported that TRPC1 and TRPC4 were overexpressed in some TNBC cell lines. For example, the Hs578T TNBC cell line showed high expression levels of TRPC1 and TRPC4 compared to the MD-MB-231 TNBC cell line, and the BT-549 TNBC cell line had high levels of TRPC4 when compared to MD-MB-231 cell line [41]. The higher TRPC expression levels in these cancer cells may render these cells more vulnerable to therapeutic strategies aimed at eliciting Ca^2+^ overload and cell death through channel agonists, rather than pharmacological inhibition to reverse hallmarks of cancer.

Support of Ca^2+^ overload being a vulnerability for cancer is that TNBC cell lines with increased TRPC4 and TRPC5 showed increased sensitivity to the TRPC4 and TRPC5 activator englerin A (EA) compared to cell lines with reduced expression [41]. Interestingly, the expression of TRPC4 and TRPC5 alone did not yield sensitivity to EA-induced and Ca^2+^-dependent cell death, indicating the importance of the heteromeric formation of TRPC members. Although TRPC1 levels in BT-549 cells were comparable to the EA-resistant cell line HCC1806, BT-549 cells showed increased sensitivity to EA-induced Ca^2+^ entry and cell death when compared to Hs578T, when TRPC4 alone is predominantly and highly expressed. A possible explanation for this is the shift of the heteromeric TRPC1/TRPC4 channel towards more Na^+^ flux [31]. In renal cell carcinoma (RCC), A498 RCC cell lines responded to englerin A by elevation of intracellular Ca^2+^ levels and followed by cell death The Ca^2+^ entry was due to activation of TRPC4 channels, as was demonstrated by patch-clamp studies. Further, while the expression of only TRPC1 did not contribute as a Ca^2+^ channel to the englerin A-induced Ca^2+^ overload and cell death, the co-expression of TRPC1 and TRPC4 and their heteromeric formation reproduced the same Ca^2+^ entry currents in HEK293 cells [58].

These findings corroborate the importance of TRPC1 expression for the overall functionality of the SOCE pathway. Our laboratory has shown that the novel cyclic peptide referred to as MTI-101 induces a robust and sustained increase in intracellular Ca^2+^ levels in multiple myeloma cells lines [78,79]. Emergence of resistance to chronic exposure of increasing concentrations of MTI-101 in the multiple myeloma H929 cell line correlated with decreased expression of the IP_3_ receptor, SERCA pump, PLCβ, TRPC1, and TRPM7. Treatment with the pharmacological inhibitor 2-ABP attenuated MTI-101-induced Ca^2+^ influx, a finding that correlated with decreased cell death. U266 and MM.1S myeloma cells with reduced expression of TRPC1 using shRNA strategies showed a reduction in MTI-101-induced cell death [78]. Further studies are required to determine the role of TRPC1 in mediating MTI-101-induced Ca^2+^ entry and whether MTI-101-induced activity is dependent on activation of TRPC1 heteromers and regulation of the SOCE pathway, thereby allowing for sustained Ca^2+^ influx and leading to caspase-independent cell death [79]. Interestingly, MTI-101 was found to be more active in primary specimens obtained from myeloma patients relapsing on standard of care agents. Together, these data suggest that stimulating Ca^2+^ overload may be a unique vulnerability to cancer cells, as experimental evidence indicates that cancer cells remodel Ca^2+^ handling pathways to favor the Ca^2+^ influx required to facilitate the hallmarks of cancer.

## 6. Clinical Outcomes

As outlined above, several TRPC members have been studied and implicated in the promotion of cancer through tumor cell proliferation, migration, invasion, and survival, as observed in lung cancer, malignant glioma, neuroblastoma, renal cell carcinoma, hepatoma, thyroid cancer, colon cancer, and breast cancer [43,44,45,46,47,57,80,81]. With respect to TRPC1, the screening of patient specimens as well as the mining of large data sets have yielded several clinically relevant insights in patients with tumors expressing TRPC1. Faouzi et al. reported TRPC1 to be expressed in all examined 17 data sets of breast cancer specimens, with no specific clinical significance [45]. Azimi et al. reported the TRPC1 expression levels in a data set from the University of North Carolina of 855 breast cancer patients, classified and reported based on the 50-signature genes subtype classification known as PAM50 [42,82]. The claudin-low breast cancer subtype exhibited the highest TRPC1 expression levels compared to other subtypes. Further, in TNBC, the mesenchymal subtype showed the highest expression level of TRPC1, and the basal subtype with lymph-node metastasis showed worsened prognoses associated with high TRPC1 expression levels [42]. In lung cancer, SOCE components including STIM1, ORAI1, and TRPC channels have been examined in a dataset of more than 2000 cases. While TRPC1 had no effect on the risk of lung cancer, two variants of TRPC4 (namely, rs9547991 and rs978156) and one variant of TRPC7 (rs11748198) were associated with increased risk of lung cancer compared to control subjects [57]. TRPC1 plasma membrane levels have been reported to be low when in inactive status, being located in close proximity to the plasma membrane. Transfer to the plasma membrane is dependent on Ca^2+^ signals generated by STIM1-ORAI1 activation [83]. Thus, membrane localization may be an important factor linking SOCE and TRPC1 functions, even if expression levels may not change. Considering the function of TRPC1, it is feasible that quantification of membrane staining of TRPC1 may lead to increased sensitivity as a prognostic indicator in cancer.

## 7. Summary

Experimental evidence indicates that although TRPC1 may not represent a driver of cancer, the expression of TRPC1 contributes to the hallmarks of cancer. To further the understanding of TRPC1 functionality in vivo, various TRPC knockout mice were created and used in different settings. A Hepta-KO cell line was created by combining cell lines with 5 TRPC knockout alleles, with TRPC2 KO, along with TRPC4 knockout, formulating a null cell line devoid of all seven TRPC members. Surprisingly, the Hepta-KO model has a functional SOCE system regardless of TRPC expression [84]. These data suggest that the SOCE can function independent of TRPC expression. However, it is feasible that TRPC1 function requires activation, such as hypoxia or a reprogramming switch to EMT, in order to modify the SOCE pathway. It is also possible that TRPC1 acts as a modulator in only a subset of cell types and/or under specific (patho)-physiological circumstances. Thus more studies are required to inform the role of TRPC1 in the context of cancer in genetically engineered mouse models. Finally, the development of more specific pharmacological tools for inhibiting or activating TRPC1 function are needed to fully validate TRPC1 as a potential target for the treatment of cancer.

## Figures and Tables

**Figure 1 cells-09-00388-f001:**
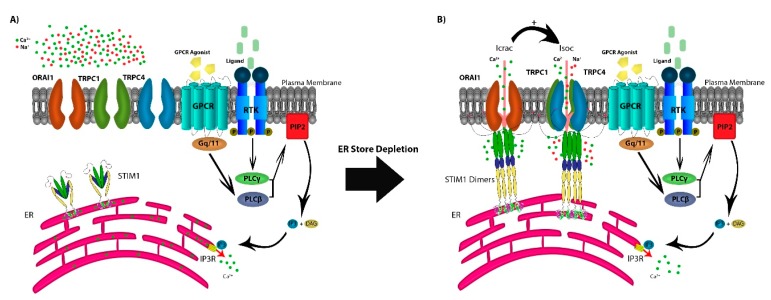
The store-operated Ca^2+^ entry pathway (SOCE). (**A**) SOCE is regulated by agonist binding to G-protein coupled receptors (GPCRs) or receptor tyrosine-kinases (RTKs), activating phospholipase Cβ (PLCβ) via Gq/11 and PLCγ via RTK-mediated signaling, resulting in the production of IP3 and DAG from the cleavage of plasma-membrane PIP2. IP3 depletes Ca^2+^ stores from the ER through the IP3R which is sensed by STIM1. (**B**) STIM molecules multimerize forming puncta and translocate to the ER–PM junction, co-assembling with the CRAC channel subunits ORAI1, activating the Ca^2+^ selective Icrac currents. Further, STIM1 forms the STIM1-ORAI1-TRPC1 complex activating cation non-selective Isoc currents.

**Figure 2 cells-09-00388-f002:**
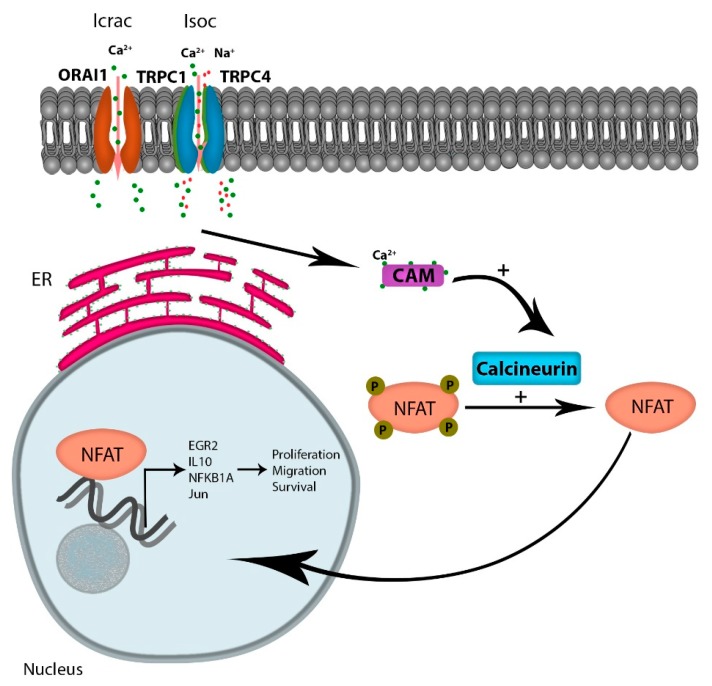
Ca^2+^ entry through SOCE activates NFAT activation: Ca^2+^ entry through the Icrac channel binds calmodulin, leading to the activation of the phosphatase protein calcineurin, activating the transcription factor NFAT. Active NFAT is translocated to the nucleus, regulating the expression of genes promoting proliferation, migration, and survival.

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
