# Peer review of "The Role of TRPC1 in Modulating Cancer Progression"

_cells, 2020, doi:10.3390/cells9020388_

Round 1

Reviewer 1 Report

This is a thoughtful review by Professor Hazlehurst research group, West Virginia University, USA. This review article includes the latest accumulated research knowledge on role of TRPC1 channel in various cancers. I have one major concern and two minor concerns, as below.

Major concern:

In order to help readers easily digest this review article, two or three graphic figures should be added.

Minor concerns:

Authors should describe the affiliation and email address in accordance with Instructions for Authors. Authors should carefully check the abbreviation the first time they use it in the text (see Instructions for Authors). Sections 4-11 should be renamed: 4-1, 4-2, 4-3, 4-4, 4-5, 5, 6, 7.

Author Response

We appreciate the thorough review of our manuscript. 

Major concern:

We now have included 2 figures and one table to help summarize information for the reader. 

Minor:

Authorship has been updated in accordance with Instructions for Authors We have renamed the sections as requested.

Reviewer 2 Report

It is a well written review on the role of TRPC1 in various cancers. The authors have clearly summarised the recent discoveries and critically evaluated them. I have only few minor concerns.

1. The subsection "TRPC1 and the hallmarks of cancer:" is redundant-adds nothing to the review in my opinion.

2. Some claims such as "Lung cancer is the leading cause of death from cancer in men and women" are without references.

3. Abbreviations were repeated e.g. EMT

4. Introduction is primarily about SOCE; can be trimmed and added with few salient findings on Ca signalling in cancer.

Author Response

We appreciate the thorough review of our manuscript

We agree, and have removed TPC1 and hallmarks of cancer As indicated, we have added a reference for the lung cancer is leading cause of cell death… We were unclear regarding repeated abbreviations we have written out the entire representation of EMT on first introduction of the term. We repeated the entire name of EMT on headings only please advise if we did not interpret this comment correctly We agree, however, considering that this review may attract cancer biologists with little background on SOCE we suggest we keep this section as is

Reviewer 3 Report

In this manuscript, the authors Elzamzamy et al. serve a review reporting the role of TRPC1 in various cancer cell types, focusing on tumor growth and metastasis. It is well written and could be very useful for the readers who are studying on TRP channels and cancer biology. I have only minor comments in Table 1, 

In the line Pancreatic, what does the ‘-SOC’ and ‘-TRPC1’ mean? If the ‘-‘ means silencing, the author should add ‘-‘ to silenced proteins in all proteins of all lines.

In the line Breast, MCF7, what does the ‘-Proliferation’ mean? I think this ‘-‘ should be removed.

Author Response

We appreciate the thorough review of our manuscript

( - ) was used to indicate “no data available”. The table was updated with bullet points, and a caption was added to the bottom of the table to indicate the dash means no data available.  

Round 2

Reviewer 1 Report

I have no more concerns.